# Compressing Word Embeddings via Deep Compositional Code Learning

**Raphael Shu**
The University of Tokyo
shu@nlab.ci.i.u-tokyo.ac.jp

**Hideki Nakayama**
The University of Tokyo
nakayama@ci.i.u-tokyo.ac.jp

## Abstract

Natural language processing (NLP) models often require a massive number of parameters for word embeddings, resulting in a large storage or memory footprint. Deploying neural NLP models to mobile devices requires compressing the word embeddings without any significant sacrifices in performance. For this purpose, we propose to construct the embeddings with few basis vectors. For each word, the composition of basis vectors is determined by a hash code. To maximize the compression rate, we adopt the multi-codebook quantization approach instead of binary coding scheme. Each code is composed of multiple discrete numbers, such as $(3, 2, 1, 8)$, where the value of each component is limited to a fixed range. We propose to directly learn the discrete codes in an end-to-end neural network by applying the Gumbel-softmax trick. Experiments show the compression rate achieves $98\%$ in a sentiment analysis task and $94\% \sim 99\%$ in machine translation tasks without performance loss. In both tasks, the proposed method can improve the model performance by slightly lowering the compression rate. Compared to other approaches such as character-level segmentation, the proposed method is language-independent and does not require modifications to the network architecture.

## 1 Introduction

Word embeddings play an important role in neural-based natural language processing (NLP) models. Neural word embeddings encapsulate the linguistic information of words in continuous vectors. However, as each word is assigned an independent embedding vector, the number of parameters in the embedding matrix can be huge. For example, when each embedding has 500 dimensions, the network has to hold 100M embedding parameters to represent 200K words. In practice, for a simple sentiment analysis model, the word embedding parameters account for 98.8% of the total parameters.

As only a small portion of the word embeddings is selected in the forward pass, the giant embedding matrix usually does not cause a speed issue. However, the massive number of parameters in the neural network results in a large storage or memory footprint. When other components of the neural network are also large, the model may fail to fit into GPU memory during training. Moreover, as the demand for low-latency neural computation for mobile platforms increases, some neural-based models are expected to run on mobile devices. Thus, it is becoming more important to compress the size of NLP models for deployment to devices with limited memory or storage capacity.

In this study, we attempt to reduce the number of parameters used in word embeddings without hurting the model performance. Neural networks are known for the significant redundancy in the connections (Denil et al., 2013). In this work, we further hypothesize that learning independent embeddings causes more redundancy in the embedding vectors, as the inter-similarity among words is ignored. Some words are very similar regarding the semantics. For example, "dog" and "dogs" have almost the same meaning, except one is plural. To efficiently represent these two words, it is desirable to share information between the two embeddings. However, a small portion in both vectors still has to be trained independently to capture the syntactic difference.

Following the intuition of creating partially shared embeddings, instead of assigning each word a unique ID, we represent each word $w$ with a code $C_w = (C_w^1, C_w^2, ..., C_w^M)$. Each component $C_w^i$ is

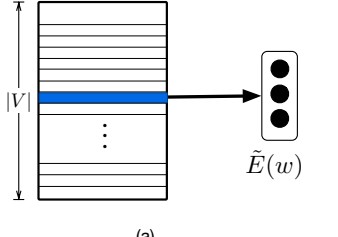 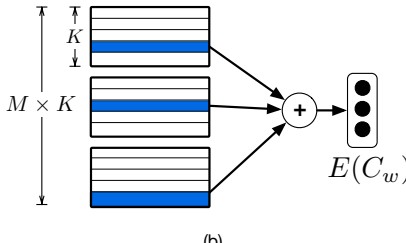

(a)  (b)

Figure 1: Comparison of embedding computations between the conventional approach (a) and compositional coding approach (b) for constructing embedding vectors

an integer number in $[1, K]$. Ideally, similar words should have similar codes. For example, we may desire $C_{\text{dog}} = (3, 2, 4, 1)$ and $C_{\text{dogs}} = (3, 2, 4, 2)$. Once we have obtained such compact codes for all words in the vocabulary, we use embedding vectors to represent the codes rather than the unique words. More specifically, we create $M$ codebooks $E_1, E_2, ..., E_M$, each containing $K$ codeword vectors. The embedding of a word is computed by summing up the codewords corresponding to all the components in the code as

$$E(C_w) = \sum_{i=1}^{M} E_i(C_w^i),  \tag{1}$$

$$\tag{2}$$

where $E_i(C_w^i)$ is the $C_w^i$-th codeword in the codebook $E_i$. In this way, the number of vectors in the embedding matrix will be $M \times K$, which is usually much smaller than the vocabulary size. Fig. 1 gives an intuitive comparison between the compositional approach and the conventional approach (assigning unique IDs). The codes of all the words can be stored in an integer matrix, denoted by $C$. Thus, the storage footprint of the embedding layer now depends on the total size of the combined codebook $E$ and the code matrix $C$.

Although the number of embedding vectors can be greatly reduced by using such coding approach, we want to prevent any serious degradation in performance compared to the models using normal embeddings. In other words, given a set of baseline word embeddings $\tilde{E}(w)$, we wish to find a set of codes $\hat{C}$ and combined codebook $\hat{E}$ that can produce the embeddings with the same effectiveness as $\tilde{E}(w)$. A safe and straight-forward way is to minimize the squared distance between the baseline embeddings and the composed embeddings as

$$(\hat{C}, \hat{E}) = \underset{C,E}{\operatorname{argmin}} \frac{1}{|V|} \sum_{w \in V} ||E(C_w) - \tilde{E}(w)||^2  \tag{3}$$

$$= \underset{C,E}{\operatorname{argmin}} \frac{1}{|V|} \sum_{w \in V} ||\sum_{i=1}^{M} E_i(C_w^i) - \tilde{E}(w)||^2 ,  \tag{4}$$

where $|V|$ is the vocabulary size. The baseline embeddings can be a set of pre-trained vectors such as word2vec (Mikolov et al., 2013) or GloVe (Pennington et al., 2014) embeddings.

In Eq. 3, the baseline embedding matrix $\tilde{E}$ is approximated by $M$ codewords selected from $M$ codebooks. The selection of codewords is controlled by the code $C_w$. Such problem of learning compact codes with multiple codebooks is formalized and discussed in the research field of compression-based source coding, known as product quantization (Jégou et al., 2011) and additive quantization (Babenko & Lempitsky, 2014; Martinez et al., 2016). Previous works learn compositional codes so as to enable an efficient similarity search of vectors. In this work, we utilize such codes for a different purpose, that is, constructing word embeddings with drastically fewer parameters.

Due to the discreteness in the hash codes, it is usually difficult to directly optimize the objective function in Eq. 3. In this paper, we propose a simple and straight-forward method to learn the codes in an end-to-end neural network. We utilize the Gumbel-softmax trick (Maddison et al., 2016;

Jang et al., 2016) to find the best discrete codes that minimize the loss. Besides the simplicity, this approach also allows one to use any arbitrary differentiable loss function, such as cosine similarity.

The contribution of this work can be summarized as follows:

- We propose to utilize the compositional coding approach for constructing the word embeddings with significantly fewer parameters. In the experiments, we show that over 98% of the embedding parameters can be eliminated in sentiment analysis task without affecting performance. In machine translation tasks, the loss-free compression rate reaches $94\% \sim 99\%$.

- We propose a direct learning approach for the codes in an end-to-end neural network, with a Gumbel-softmax layer to encourage the discreteness.

- The neural network for learning codes will be packaged into a tool[1]. With the learned codes and basis vectors, the computation graph for composing embeddings is fairly easy to implement, and does not require modifications to other parts in the neural network.

## 2 RELATED WORK

Existing works for compressing neural networks include low-precision computation (Vanhoucke et al., 2011; Hwang & Sung, 2014; Courbariaux et al., 2014; Anwar et al., 2015), quantization (Chen et al., 2015; Han et al., 2016; Zhou et al., 2017), network pruning (LeCun et al., 1989; Hassibi & Stork, 1992; Han et al., 2015; Wen et al., 2016) and knowledge distillation (Hinton et al., 2015). Network quantization such as HashedNet (Chen et al., 2015) forces the weight matrix to have few real weights, with a hash function to determine the weight assignment. To capture the non-uniform nature of the networks, DeepCompression (Han et al., 2016) groups weight values into clusters based on pre-trained weight matrices. The weight assignment for each value is stored in the form of Huffman codes. However, as the embedding matrix is tremendously big, the number of hash codes a model need to maintain is still large even with Huffman coding.

Network pruning works in a different way that makes a network sparse. Iterative pruning (Han et al., 2015) prunes a weight value if its absolute value is smaller than a threshold. The remaining network weights are retrained after pruning. Some recent works (See et al., 2016; Zhang et al., 2017) also apply iterative pruning to prune 80% of the connections for neural machine translation models. In this paper, we compare the proposed method with iterative pruning.

The problem of learning compact codes considered in this paper is closely related to learning to hash (Weiss et al., 2008; Kulis & Darrell, 2009; Liu et al., 2012), which aims to learn the hash codes for vectors to facilitate the approximate nearest neighbor search. Initiated by product quantization (Jégou et al., 2011), subsequent works such as additive quantization (Babenko & Lempitsky, 2014) explore the use of multiple codebooks for source coding, resulting in compositional codes. We also adopt the coding scheme of additive quantization for its storage efficiency. Previous works mainly focus on performing efficient similarity search of image descriptors. In this work, we put more focus on reducing the codebook sizes and learning efficient codes to avoid performance loss. Joulin et al. (2016) utilizes an improved version of product quantization to compress text classification models. However, to match the baseline performance, much longer hash codes are required by product quantization. This will be detailed in Section 5.2. Concurrent to this work, Chen et al. (2017) also explores the similar idea and obtained positive results in language modeling tasks. Also, Raunak (2017) tried to reduce dimension of embeddings using PCA.

To learn the codebooks and code assignment, additive quantization alternatively optimizes the codebooks and the discrete codes. The learning of code assignment is performed by Beam Search algorithm when the codebooks are fixed. In this work, we propose a straight-forward method to directly learn the code assignment and codebooks simutaneously in an end-to-end neural network.

Some recent works (Xia et al., 2014; Liu et al., 2016; Yang et al., 2017) in learning to hash also utilize neural networks to produce binary codes by applying binary constrains (e.g., sigmoid function). In this work, we encourage the discreteness with the Gumbel-Softmax trick for producing compositional codes.

---

[1]The code can be found in https://github.com/zomux/neuralcompressor

As an alternative to our approach, one can also reduce the number of unique word types by forcing a character-level segmentation. Kim et al. (2016) proposed a character-based neural language model, which applies a convolutional layer after the character embeddings. Botha et al. (2017) propose to use char-gram as input features, which are further hashed to save space. Generally, using character-level inputs requires modifications to the model architecture. Moreover, some Asian languages such as Japanese and Chinese retain a large vocabulary at the character level, which makes the character-based approach difficult to be applied. In contrast, our approach does not suffer from these limitations.

## 3 ADVANTAGE OF COMPOSITIONAL CODES

In this section, we formally describe the compositional coding approach and analyze its merits for compressing word embeddings. The coding approach follows the scheme in additive quantization (Babenko & Lempitsky, 2014). We represent each word $w$ with a compact code $C_w$ that is composed of $M$ components such that $C_w \in Z_+^M$. Each component $C_w^i$ is constrained to have a value in $[1, K]$, which also indicates that $M \log_2 K$ bits are required to store each code. For convenience, $K$ is selected to be a number of a multiple of 2, so that the codes can be efficiently stored.

If we restrict each component $C_w^i$ to values of 0 or 1, the code for each word $C_w$ will be a binary code. In this case, the code learning problem is equivalent to a matrix factorization problem with binary components. Forcing the compact codes to be binary numbers can be beneficial, as the learning problem is usually easier to solve in the binary case, and some existing optimization algorithms in learning to hash can be reused. However, the compositional coding approach produces shorter codes and is thus more storage efficient.

As the number of basis vectors is $M \times K$ regardless of the vocabulary size, the only uncertain factor contributing to the model size is the size of the hash codes, which is proportional to the vocabulary size. Therefore, maintaining short codes is cruicial in our work. Suppose we wish the model to have a set of $N$ basis vectors. Then in the binary case, each code will have $N/2$ bits. For the compositional coding approach, if we can find a $M \times K$ decomposition such that $M \times K = N$, then each code will have $M \log_2 K$ bits. For example, a binary code will have a length of 256 bits to support 512 basis vectors. In contrast, a $32 \times 16$ compositional coding scheme will produce codes of only 128 bits.

|  | #vectors | computation | code length (bits) |
|---|---|---|---|
| conventional | $|V|$ | 1 | - |
| binary | $N$ | $N/2$ | $N/2$ |
| compositional | $MK$ | $M$ | $M \log_2 K$ |

Table 1: Comparison of different coding approaches. To support $N$ basis vectors, a binary code will have $N/2$ bits and the embedding computation is a summation over $N/2$ vectors. For the compositional approach with $M$ codebooks and $K$ codewords in each codebook, each code has $M \log_2 K$ bits, and the computation is a summation over $M$ vectors.

A comparison of different coding approaches is summarized in Table 1. We also report the number of basis vectors required to compute an embedding as a measure of computational cost. For the conventional approach, the number of vectors is identical to the vocabulary size and the computation is basically a single indexing operation. In the case of binary codes, the computation for constructing an embedding involves a summation over $N/2$ basis vectors. For the compositional approach, the number of vectors required to construct an embedding vector is $M$. Both the binary and compositional approaches have significantly fewer vectors in the embedding matrix. The compositional coding approach provides a better balance with shorter codes and lower computational cost.

## 4 CODE LEARNING WITH GUMBEL-SOFTMAX

Let $\tilde{\mathbf{E}} \in \mathbb{R}^{|V| \times H}$ be the original embedding matrix, where each embedding vector has $H$ dimensions. By using the reconstruction loss as the objective function in Eq. 3, we are actually finding

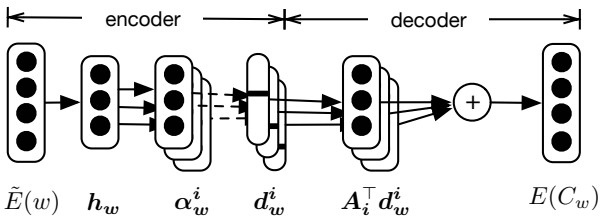

Figure 2: The network architecture for learning compositional compact codes. The Gumbel-softmax computation is marked with dashed lines.

an approximate matrix factorization $\tilde{\mathbf{E}} \approx \sum_{i=0}^{M} \boldsymbol{D^i} \boldsymbol{A_i}$, where $\boldsymbol{A_i} \in \mathbb{R}^{K \times H}$ is a basis matrix for the $i$-th component. $\boldsymbol{D^i}$ is a $|V| \times K$ code matrix in which each row is an $K$-dimensional one-hot vector. If we let $\boldsymbol{d_w^i}$ be the one-hot vector corresponding to the code component $C_w^i$ for word $w$, the computation of the word embeddings can be reformulated as

$$E(C_w) = \sum_{i=0}^{M} \boldsymbol{A_i^\top} \boldsymbol{d_w^i}. \tag{5}$$

Therefore, the problem of learning discrete codes $C_w$ can be converted to a problem of finding a set of optimal one-hot vectors $\boldsymbol{d_w^1}, ..., \boldsymbol{d_w^M}$ and source dictionaries $\boldsymbol{A_1}, ..., \boldsymbol{A_M}$, that minimize the reconstruction loss. The Gumbel-softmax reparameterization trick (Maddison et al., 2016; Jang et al., 2016) is useful for parameterizing a discrete distribution such as the $K$-dimensional one-hot vectors $\boldsymbol{d_w^i}$ in Eq. 5. By applying the Gumbel-softmax trick, the $k$-th elemement in $\boldsymbol{d_w^i}$ is computed as

$$\left(\boldsymbol{d_w^i}\right)_k = \text{softmax}_\tau (\log \boldsymbol{\alpha_w^i} + G)_k \tag{6}$$

$$= \frac{\exp((\log (\boldsymbol{\alpha_w^i})_k + G_k)/\tau)}{\sum_{k'=1}^{K} \exp((\log (\boldsymbol{\alpha_w^i})_{k'} + G_{k'})/\tau)} , \tag{7}$$

where $G_k$ is a noise term that is sampled from the Gumbel distribution $-\log(-\log(\text{Uniform}[0,1]))$, whereas $\tau$ is the temperature of the softmax. In our model, the vector $\boldsymbol{\alpha_w^i}$ is computed by a simple neural network with a single hidden layer as

$$\boldsymbol{\alpha_w^i} = \text{softplus}(\boldsymbol{\theta_i'}^\top \boldsymbol{h_w} + \boldsymbol{b_i'}), \tag{8}$$

$$\boldsymbol{h_w} = \tanh(\boldsymbol{\theta}^\top \tilde{E}(w) + \boldsymbol{b}) . \tag{9}$$

In our experiments, the hidden layer $\boldsymbol{h_w}$ always has a size of $MK/2$. We found that a fixed temperature of $\tau = 1$ just works well. The Gumbel-softmax trick is applied to $\boldsymbol{\alpha_w^i}$ to obtain $\boldsymbol{d_w^i}$. Then, the model reconstructs the embedding $E(C_w)$ with Eq. 5 and computes the reconstruction loss with Eq. 3. The model architecture of the end-to-end neural network is illustrated in Fig. 2, which is effectively an auto-encoder with a Gumbel-softmax middle layer. The whole neural network for coding learning has five parameters $(\boldsymbol{\theta}, \boldsymbol{b}, \boldsymbol{\theta'}, \boldsymbol{b'}, \boldsymbol{A})$.

Once the coding learning model is trained, the code $C_w$ for each word can be easily obtained by applying $\text{argmax}$ to the one-hot vectors $\boldsymbol{d_w^1}, ..., \boldsymbol{d_w^M}$. The basis vectors (codewords) for composing the embeddings can be found as the row vectors in the weight matrix $\boldsymbol{A}$.

For general NLP tasks, one can learn the compositional codes from publicly available word vectors such as GloVe vectors. However, for some tasks such as machine translation, the word embeddings are usually jointly learned with other parts of the neural network. For such tasks, one has to first train a normal model to obtain the baseline embeddings. Then, based on the trained embedding matrix, one can learn a set of task-specific codes. As the reconstructed embeddings $E(C_w)$ are not identical to the original embeddings $\tilde{E}(w)$, the model parameters other than the embedding matrix have to be retrained again. The code learning model cannot be jointly trained with the machine translation model as it takes far more iterations for the coding layer to converge to one-hot vectors.

## 5 EXPERIMENTS

In our experiments, we focus on evaluating the maximum loss-free compression rate of word embeddings on two typical NLP tasks: sentiment analysis and machine translation. We compare the model performance and the size of embedding layer with the baseline model and the iterative pruning method (Han et al., 2015). Please note that the sizes of other parts in the neural networks are not included in our results. For dense matrices, we report the size of dumped numpy arrays. For the sparse matrices, we report the size of dumped *compressed sparse column matrices* (csc_matrix) in scipy. All float numbers take 32 bits storage. We enable the "compressed" option when dumping the matrices, without this option, the file size is about 1.1 times bigger.

### 5.1 CODE LEARNING

To learn efficient compact codes for each word, our proposed method requires a set of baseline embedding vectors. For the sentiment analysis task, we learn the codes based on the publicly available GloVe vectors. For the machine translation task, we first train a normal neural machine translation (NMT) model to obtain task-specific word embeddings. Then we learn the codes using the pre-trained embeddings.

We train the end-to-end network described in Section 4 to learn the codes automatically. In each iteration, a small batch of the embeddings is sampled uniformly from the baseline embedding matrix. The network parameters are optimized to minimize the reconstruction loss of the sampled embeddings. In our experiments, the batch size is set to 128. We use Adam optimizer (Kingma & Ba, 2014) with a fixed learning rate of 0.0001. The training is run for 200K iterations. Every 1,000 iterations, we examine the loss on a fixed validation set and save the parameters if the loss decreases. We evenly distribute the model training to 4 GPUs using the *nccl* package, so that one round of code learning takes around 15 minutes to complete.

### 5.2 SENTIMENT ANALYSIS

**Dataset:** For sentiment analysis, we use a standard separation of IMDB movie review dataset (Maas et al., 2011), which contains 25k reviews for training and 25K reviews for testing purpose. We lowercase and tokenize all texts with the *nltk* package. We choose the 300-dimensional uncased GloVe word vectors (trained on 42B tokens of Common Crawl data) as our baseline embeddings. The vocabulary for the model training contains all words appears both in the IMDB dataset and the GloVe vocabulary, which results in around 75K words. We truncate the texts of reviews to assure they are not longer than 400 words.

**Model architecture:** Both the baseline model and the compressed models have the same computational graph except the embedding layer. The model is composed of a single LSTM layer with 150 hidden units and a softmax layer for predicting the binary label. For the baseline model, the embedding layer contains a large $75K \times 300$ embedding matrix initialized by GloVe embeddings. For the compressed models based on the compositional coding, the embedding layer maintains a matrix of basis vectors. Suppose we use a $32 \times 16$ coding scheme, the basis matrix will then have a shape of $512 \times 300$, which is initialized by the concatenated weight matrices $[\boldsymbol{A_1}; \boldsymbol{A_2}; ...; \boldsymbol{A_M}]$ in the code learning model. The embedding parameters for both models remain fixed during the training. For the models with network pruning, the sparse embedding matrix is finetuned during training.

**Training details:** The models are trained with Adam optimizer for 15 epochs with a fixed learning rate of 0.0001. At the end of each epoch, we evaluate the loss on a small validation set. The parameters with lowest validation loss are saved.

**Results:** For different settings of the number of components $M$ and the number of codewords $K$, we train the code learning network. The average reconstruction loss on a fixed validation set is summarized in the left of Table 2. For reference, we also report the total size (MB) of the embedding layer in the right table, which includes the sizes of the basis matrix and the hash table. We can see that increasing either $M$ or $K$ can effectively decrease the reconstruction loss. However, setting $M$ to a large number will result in longer hash codes, thus significantly increase the size of the embedding layer. Hence, it is important to choose correct numbers for $M$ and $K$ to balance the performance and model size.

| loss | M=8 | M=16 | M=32 | M=64 | | size (MB) | M=8 | M=16 | M=32 | M=64 |
|------|-----|------|------|------|--|-----------|-----|------|------|------|
| K=8  | 29.1 | 25.8 | 21.9 | 15.5 | | K=8  | 0.28 | 0.56 | 1.12 | 2.24 |
| K=16 | 27.0 | 22.8 | 19.1 | 11.5 | | K=16 | 0.41 | 0.83 | 1.67 | 3.34 |
| K=32 | 24.4 | 20.4 | 14.3 | 9.3  | | K=32 | 0.62 | 1.24 | 2.48 | 4.96 |
| K=64 | 21.9 | 16.9 | 12.1 | 7.6  | | K=64 | 0.95 | 1.91 | 3.82 | 7.64 |

Table 2: Reconstruction loss and the size of embedding layer (MB) of difference settings

To see how the reconstructed loss translates to the classification accuracy, we train the sentiment analysis model for different settings of code schemes and report the results in Table 3. The baseline model using 75k GloVe embeddings achieves an accuracy of 87.18 with an embedding matrix using 78 MB of storage. In this task, forcing a high compression rate with iterative pruning degrades the classification accuracy.

| | #vectors | vector size | code len | code size | total size | accuracy |
|--|----------|-------------|----------|-----------|------------|----------|
| GloVe baseline | 75102 | 78 MB | - | - | 78 MB | 87.18 |
| prune 80% | 75102 | 21 MB | - | - | 21 MB | 86.25 |
| prune 90% | 75102 | 11 MB | - | - | 11 MB | 84.96 |
| NPQ ($10 \times 256$) | 256 | 0.26 MB | 80 bits | 0.71 MB | 0.97 MB | 86.21 |
| NPQ ($60 \times 256$) | 256 | 0.26 MB | 480 bits | 4.26 MB | 4.52 MB | 87.11 |
| $8 \times 64$ coding | 512 | 0.52 MB | 48 bits | 0.42 MB | 0.94 MB | 86.66 |
| $16 \times 32$ coding | 512 | 0.52 MB | 80 bits | 0.71 MB | **1.23 MB** | 87.37 |
| $32 \times 16$ coding | 512 | 0.52 MB | 128 bits | 1.14 MB | 1.66 MB | 87.80 |
| $64 \times 8$ coding | 512 | 0.52 MB | 192 bits | 1.71 MB | 2.23 MB | **88.15** |

Table 3: Trade-off between the model performance and the size of embedding layer on IMDB sentiment analysis task

We also show the results using normalized product quantization (NPQ) (Joulin et al., 2016). We quantize the filtered GloVe embeddings with the codes provided by the authors, and train the models based on the quantized embeddings. To make the results comparable, we report the codebook size in numpy format. For our proposed methods, the maximum loss-free compression rate is achieved by a $16 \times 32$ coding scheme. In this case, the total size of the embedding layer is 1.23 MB, which is equivalent to a compression rate of 98.4%. We also found the classification accuracy can be substantially improved with a slightly lower compression rate. The improved model performance may be a byproduct of the strong regularization.

## 5.3 Machine Translation

**Dataset:** For machine translation tasks, we experiment on IWSLT 2014 German-to-English translation task (Cettolo et al., 2014) and ASPEC English-to-Japanese translation task (Nakazawa et al., 2016). The IWSLT14 training data contains 178K sentence pairs, which is a small dataset for machine translation. We utilize moses toolkit (Koehn et al., 2007) to tokenize and lowercase both sides of the texts. Then we concatenate all five TED/TEDx development and test corpus to form a test set containing 6750 sentence pairs. We apply byte-pair encoding (Sennrich et al., 2016) to transform the texts to subword level so that the vocabulary has a size of 20K for each language. For evaluation, we report *tokenized BLEU* using "multi-bleu.perl".

The ASPEC dataset contains 300M bilingual pairs in the training data with the automatically estimated quality scores provided for each pair. We only use the first 150M pairs for training the models. The English texts are tokenized by moses toolkit whereas the Japanese texts are tokenized by kytea (Neubig et al., 2011). The vocabulary size for each language is reduced to 40K using byte-pair encoding. The evaluation is performed using a standard kytea-based post-processing script for this dataset.

**Model architecture:** In our preliminary experiments, we found a $32 \times 16$ coding works well for a vanilla NMT model. As it is more meaningful to test on a high-performance model, we applied several techniques to improve the performance. The model has a standard bi-directional encoder

composed of two LSTM layers similar to Bahdanau et al. (2015). The decoder contains two LSTM layers. Residual connection (He et al., 2016) with a scaling factor of $\sqrt{1/2}$ is applied to the two decoder states to compute the outputs. All LSTMs and embeddings have 256 hidden units in the IWSLT14 task and 1000 hidden units in ASPEC task. The decoder states are firstly linearly transformed to 600-dimensional vectors before computing the final softmax. Dropout with a rate of 0.2 is applied everywhere except the recurrent computation. We apply Key-Value Attention (Miller et al., 2016) to the first decoder, where the query is the sum of the feedback embedding and the previous decoder state and the keys are computed by linear transformation of encoder states.

**Training details:** All models are trained by Nesterov's accelerated gradient (Nesterov, 1983) with an initial learning rate of 0.25. We evaluate the smoothed BLEU (Lin & Och, 2004) on a validation set composed of 50 batches every 7,000 iterations. The learning rate is reduced by a factor of 10 if no improvement is observed in 3 validation runs. The training ends after the learning rate is reduced three times. Similar to the code learning, the training is distributed to 4 GPUs, each GPU computes a mini-batch of 16 samples.

We firstly train a baseline NMT model to obtain the task-specific embeddings for all in-vocabulary words in both languages. Then based on these baseline embeddings, we obtain the hash codes and basis vectors by training the code learning model. Finally, the NMT models using compositional coding are retrained by plugging in the reconstructed embeddings. Note that the embedding layer is fixed in this phase, other parameters are retrained from random initial values.

**Results:** The experimental results are summarized in Table 4. All translations are decoded by the beam search with a beam size of 5. The performance of iterative pruning varies between tasks. The loss-free compression rate reaches 92% on ASPEC dataset by pruning 90% of the connections. However, with the same pruning ratio, a modest performance loss is observed in IWSLT14 dataset.

For the models using compositional coding, the loss-free compression rate is 94% for the IWSLT14 dataset and 99% for the ASPEC dataset. Similar to the sentiment analysis task, a significant performance improvement can be observed by slightly lowering the compression rate. Note that the sizes of NMT models are still quite large due to the big softmax layer and the recurrent layers, which are not reported in the table. Please refer to existing works such as Zhang et al. (2017) for the techniques of compressing layers other than word embeddings.

|  | coding | #vectors | vector size | code len | code size | total size | BLEU(%) |
|---|---|---|---|---|---|---|---|
| | baseline | 40000 | 35 MB | - | - | 35 MB | 29.45 |
| | prune 90% | 40000 | 5.21 MB | - | - | 5.21 MB | 29.34 |
| De → En | prune 95% | 40000 | 2.63 MB | - | - | 2.63 MB | 28.84 |
| | $32 \times 16$ | 512 | 0.44 MB | 128 bits | 0.61 MB | 1.05 MB | 29.04 |
| | $64 \times 16$ | 1024 | 0.89 MB | 256 bits | 1.22 MB | **2.11 MB** | **29.56** |
| | baseline | 80000 | 274 MB | - | - | 274 MB | 37.93 |
| | prune 90% | 80000 | 41 MB | - | - | 41 MB | 38.56 |
| En → Ja | prune 98% | 80000 | 8.26 MB | - | - | 8.26 MB | 37.09 |
| | $32 \times 16$ | 512 | 1.75 MB | 128 bits | 1.22 MB | **2.97 MB** | 38.10 |
| | $64 \times 16$ | 1024 | 3.50 MB | 256 bits | 2.44 MB | 5.94 MB | **38.89** |

Table 4: Trade-off between the model performance and the size of embedding layer in machine translation tasks

# 6 QUALITATIVE ANALYSIS

## 6.1 EXAMPLES OF LEARNED CODES

In Table 5, we show some examples of learned codes based on the 300-dimensional uncased GloVe embeddings used in the sentiment analysis task. We can see that the model learned to assign similar codes to the words with similar meanings. Such a code-sharing mechanism can significantly reduce the redundancy of the word embeddings, thus helping to achieve a high compression rate.

| category | word | 8 × 8 **code** | | | | | | | | 16 × 16 **code** | | | | | | | | | | | | | | | |
|---|---|---|---|---|---|---|---|---|---|---|---|---|---|---|---|---|---|---|---|---|---|---|---|---|---|
| animal | dog | 0 | 7 | 0 | 1 | 7 | 3 | 7 | 0 | 7 | 7 | 0 | 8 | 3 | 5 | 8 | 5 | B | 2 | E | E | 0 | B | 0 | A |
| | cat | 7 | 7 | 0 | 1 | 7 | 3 | 7 | 0 | 7 | 7 | 2 | 8 | B | 5 | 8 | C | B | 2 | E | E | 4 | B | 0 | A |
| | penguin | 0 | 7 | 0 | 1 | 7 | 3 | 6 | 0 | 7 | 7 | E | 8 | 7 | 6 | 4 | C | F | D | E | 3 | D | 8 | 0 | A |
| verb | go | 7 | 7 | 0 | 6 | 4 | 3 | 3 | 0 | 2 | C | C | 8 | 2 | C | 1 | 1 | B | D | 0 | E | 0 | B | 5 | 8 |
| | went | 4 | 0 | 7 | 6 | 4 | 3 | 2 | 0 | B | C | C | 6 | B | C | 7 | 5 | B | 8 | 6 | E | 0 | D | 0 | 4 |
| | gone | 7 | 7 | 0 | 6 | 4 | 3 | 3 | 0 | 2 | C | C | 8 | 0 | B | 1 | 5 | B | D | 6 | E | 0 | 2 | 5 | A |

Table 5: Examples of learned compositional codes based on GloVe embedding vectors

## 6.2 ANALYSIS OF CODE EFFICIENCY

Besides the performance, we also care about the storage efficiency of the codes. In the ideal situation, all codewords shall be fully utilized to convey a fraction of meaning. However, as the codes are automatically learned, it is possible that some codewords are abandoned during the training. In extreme cases, some "dead" codewords can be used by none of the words.

To analyze the code efficiency, we count the number of words that contain a specific subcode in each component. Figure 3 gives a visualization of the code balance for three coding schemes. Each column shows the counts of the subcodes of a specific component. In our experiments, when using a 8 × 8 coding scheme, we found 31% of the words have a subcode "0" for the first component, while the subcode "1" is only used by 5% of the words. The assignment of codes is more balanced for larger coding schemes. In any coding scheme, even the most unpopular codeword is used by about 1000 words. This result indicates that the code learning model is capable of assigning codes efficiently without wasting a codeword.

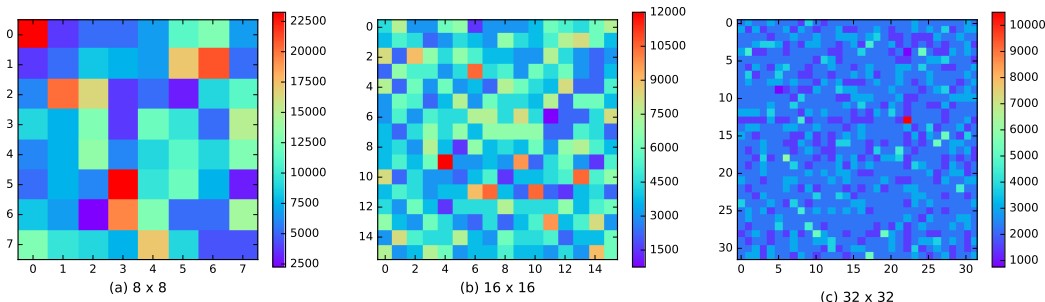

Figure 3: Visualization of code balance for different coding scheme. Each cell in the heat map shows the count of words containing a specific subcode. The results show that any codeword is assigned to more than 1000 words without wasting.

## 7 CONCLUSION

In this work, we propose a novel method for reducing the number of parameters required in word embeddings. Instead of assigning each unique word an embedding vector, we compose the embedding vectors using a small set of basis vectors. The selection of basis vectors is governed by the hash code of each word. We apply the compositional coding approach to maximize the storage efficiency. The proposed method works by eliminating the redundancy inherent in representing similar words with independent embeddings. In our work, we propose a simple way to directly learn the discrete codes in a neural network with Gumbel-softmax trick. The results show that the size of the embedding layer was reduced by 98% in IMDB sentiment analysis task and $94\% \sim 99\%$ in machine translation tasks without affecting the performance.

Our approach achieves a high loss-free compression rate by considering the semantic inter-similarity among different words. In qualitative analysis, we found the learned codes of similar words are very close in Hamming space. As our approach maintains a dense basis matrix, it has the potential

to be further compressed by applying pruning techniques to the dense matrix. The advantage of compositional coding approach will be more significant if the size of embedding layer is dominated by the hash codes.

## ACKNOWLEDGMENTS

This work is supported by JSPS KAKENHI Grant Number 16H05872.

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

## A    APPENDIX: SHARED CODES

In both tasks, when we use a small code decomposition, we found some hash codes are assigned to multiple words. Table 6 lists some samples of shared codes with their corresponding words from the sentiment analysis task. This phenomenon does not cause a problem in either task, as the words only have shared codes when they have almost the same sentiments or target translations.

| shared code | words |
|---|---|
| 4 7 7 0 4 7 1 1 | homes cruises motel hotel resorts mall vacations hotels |
| 6 6 7 1 4 0 2 0 | basketball softball nfl nascar baseball defensive ncaa tackle nba |
| 3 7 3 2 4 3 3 0 | unfortunately hardly obviously enough supposed seem totally ... |
| 4 6 7 0 4 7 5 0 | toronto oakland phoenix miami sacramento denver minneapolis ... |
| 7 7 6 6 7 3 0 0 | yo ya dig lol dat lil bye |

Table 6: Examples of words sharing same codes when using a $8 \times 8$ code decomposition

## B    APPENDIX: SEMANTICS OF CODES

In order to see whether each component captures semantic meaning. we learned a set of codes using a 3 x 256 coding scheme, this will force the model to decompose each embedding into 3 vectors. In order to maximize the compression rate, the model must make these 3 vectors as independent as possible.

| word | code |
|---|---|
| man | 210 153 153 |
| woman | 232 153 153 |
| king | 210 180 039 |
| queen | 232 180 039 |
| British | 118 132 142 |
| London | 185 126 142 |
| Japan | 118 056 021 |
| Tokyo | 185 036 021 |

Table 7: Some code examples using a $3 \times 256$ coding scheme.

As we can see from Table 7, we can transform "man/king" to "woman/queen" by change the subcode "210" in the first component to "232". So we can think "210" must be a "male" code, and "232" must be a "female" code. Such phenomenon can also be observed in other words such as city names.

