# OpenReview forum: "Compressing Word Embeddings via Deep Compositional Code Learning"
_ICLR.cc/2018/Conference — Accept (Poster)_

### Official Review · AnonReviewer1 · 2017-11-27
**Effective work**

**Rating:** 8
**Confidence:** 4

**Review:**

This paper proposed a new method to compress the space complexity of word embedding vectors by introducing summation composition over a limited number of basis vectors, and representing each embedding as a list of the basis indices. The proposed method can reduce more than 90% memory consumption while keeping original model accuracy in both the sentiment analysis task and the machine translation tasks.

Overall, the paper is well-written. The motivation is clear, the idea and approaches look suitable and the results clearly follow the motivation.

I think it is better to clarify in the paper that the proposed method can reduce only the complexity of the input embedding layer. For example, the model does not guarantee to be able to convert resulting "indices" to actual words (i.e., there are multiple words that have completely same indices, such as rows 4 and 6 in Table 5), and also there is no trivial method to restore the original indices from the composite vector. As a result, the model couldn't be used also as the proxy of the word prediction (softmax) layer, which is another but usually more critical bottleneck of the machine translation task.
For reader's comprehension, it would like to add results about whole memory consumption of each model as well.
Also, although this paper is focused on only the input embeddings, authors should refer some recent papers that tackle to reduce the complexity of the softmax layer. There are also many studies, and citing similar approaches may help readers to comprehend overall region of these studies.

Furthermore, I would like to see two additional analysis. First, if we trained the proposed model with starting from "zero" (e.g., randomly settling each index value), what results are obtained? Second, What kind of information is distributed in each trained basis vector? Are there any common/different things between bases trained by different tasks?

---

> ### Public Comment · (anonymous) · 2017-12-04
> **Response for AnonReviewer1**
>
> Thank you for spending the time to review our paper. Hope our response can answer your question.
>
> 1. About compressing the Softmax layer
>
> We spent a significant period of time trying to apply the proposed method to the softmax layer, though without a successful result. It can be caused by the code sharing problem (multiple words get the same code) or the loss function. However, we are still optimistic that the compositional coding approach can also be applied to the softmax, which should be our future work. We will refer the readers to some recent papers that reduce the size of the softmax layer using pruning techniques.
>
> 2. About the total model size
>
> The full sizes of the baseline models are summarized in the following table:
>
> Task                          Embed      Full size   Ratio of embed
> IMDB                         78 MB        79.1 MB         98.6%
> IWSLT De-En           35 MB         94   MB         37.2%
> ASPEC En-Ja            274 MB       506 MB         54.1%
>
> We will put the information of full model sizes into the tables in the experiment section.
>
> 3. Experiment with random code assignment
>
> We tried to initialize a set of 32 x 16 codes to be random numbers and see the performance in the IMDB task. With the random code assignment, the accuracy is much lower than the baseline.
>
> ------------------------------------------------------------------------------------
> Model                                                            Accuracy
> Baseline                                                        87.18
> Random code + trained codebooks           84.19
> Random code + random codebooks          84.72
> ---------------------------------------------------------------
>
> 4. Analysis of information distributed in the codes
>
> As the codes are learned by a neural net, the interpretability is not guaranteed. However, we found some interesting relations in the codes.
>
> For animal names, the 3rd subcode is normally a "5" for the plural nouns. For the verbs, we found the 2nd subcode is normally a "0" if the verb is in the past tense. Although there are also violations, we believe the model learned to arrange the codes in an efficient way.
>
> ----
> dog        7 7 0 1 7 3 7 0
> dogs      4 7 5 1 7 3 4 0
>
> cat        0 7 0 1 7 3 7 0
> cats      4 7 5 1 7 3 4 0
>
> pig        7 3 6 1 7 3 4 7
> pigs      7 3 5 1 7 3 4 0
>
> fish       7 7 6 1 4 3 4 7
> fishes    7 2 5 0 7 3 4 6
>
> fox        6 5 7 1 4 3 0 0
> foxes    6 2 5 1 7 3 4 6
> ----
> buy          0 7 2 1 4 3 3 1
> bought    0 0 2 1 4 3 3 1
>
> kick        7 6 1 1 4 3 0 0
> kicked    7 0 1 1 4 3 2 0
>
> go         7 7 0 6 4 3 3 0
> went    4 0 7 6 4 3 2 0
>
> pick        7 6 7 1 4 3 3 0
> picked    7 0 7 1 4 0 3 0
>
> catch       7 7 1 6 4 3 6 0
> caught     7 0 7 4 4 3 2 0
> ----

---

### Official Review · AnonReviewer2 · 2017-12-01

**Rating:** 6
**Confidence:** 4

**Review:**

This paper presents an interesting idea to word embeddings that it combines a few base vectors to generate new word embeddings. It also adopts an interesting multicodebook approach for encoding than binary embeddings.

The paper presents the proposed approach to a few NLP problems and have shown that this is able to significant reduce the size, increase compression ratio, and still achieved good accuracy.

The experiments are convincing and solid. Overall I am weakly inclined to accept this paper.

---

### Official Review · AnonReviewer3 · 2017-12-04
**Simple yet effective work**

**Rating:** 7
**Confidence:** 4

**Review:**

The authors proposed to compress word embeddings by approximate matrix factorization, and to solve the problem with the Gumbel-soft trick. The proposed method achieved compression rate 98% in a sentiment analysis task, and compression rate over 94% in machine translation tasks, without a performance loss.

This paper is well-written and easy to follow.  The motivation is clear and the idea is simple and effective.

It would be better to provide deeper analysis in Subsection 6.1. The current analysis is too simple. It may be interesting to explain the meanings of individual components. Does each component is related to a certain topic? Is it meaningful to perform ADD or SUBSTRACT on the leaned code?

It may also be interesting to provide suitable theoretical analysis, e.g., relationships with the SVD of the embedding matrix.

---

> ### Public Comment · (anonymous) · 2017-12-05
> **Response for AnonReviewer3**
>
> Thank you for spending the time to review our paper.
>
> As multiple reviewers are asking for us to analyze the information learned by each component, we did some extra experiments and found some interesting results.
>
> We tried to learn a set of codes using a 3 x 256 coding scheme, this will force the model to decompose each embedding into 3 vectors. In order to maximize the compression rate, the model must make these 3 vectors as independent as possible. So we can think that they represent 3 concepts.
>
> Then we extracted the codes of some related words:
> ------------------------------
> man	    210 153 153
> woman	    232 153 153
>
> king	     210 180 39
> queen	     232 180 39
>
> british	    118 132 142
> london	    185 126 142
>
> japan	    118 56 21
> tokyo	    185 36 21
> ------------------------------
>
> We can transform a "man/king" to "woman/queen" by change the subcode "210" in the first component to "232".
> So we can think "210" must be a "male" code, and "232" must be a "female" code.
>
> Similarly, when we look at the country and city names, we can find "185" in the first component to be a "city" code.
>
> We uploaded the 3x256 and 8x8 codes of 10,000 most frequent words to anonymous gists, so the those who are interested in the codes can have a look.
>
> ------
> 3 x 256 codes of 10k most frequent words:
> https://gist.github.com/anonymous/aa6c03f871900a3c4e5d7f65d74361fe
>
> 8 x 8 codes of 10k most frequent words:
> https://gist.github.com/anonymous/584d64a28c3bb7c421eee8450cae823a

---

### Public Comment · (anonymous) · 2017-12-04
**Comparaison with related word embeddings compression methods and literature**

I am curious to know if this method performs better than existing word embeddings compression techniques such as Product Quantizers (which also exploits the idea of compositional codes) [1] or WTA autoencoders [2].

[1] FastText.zip: Compressing text classification models https://arxiv.org/pdf/1612.03651.pdf
[2] ANDREWS, Martin. Compressing word embeddings. In : International Conference on Neural Information Processing. Springer International Publishing, 2016. p. 413-422.

---

> ### Public Comment · (anonymous) · 2017-12-05
> **Response to Public Comment 1**
>
> Actually, we just obtained the codes from the authors of FastText.zip and finished the comparison a few weeks ago.
>
> Their idea is based on normalized product quantization (NPQ), which split a vector into K parts and quantize each part. For each word, an extra byte is used to quantize the norm of the embedding vector. We found one drawback of this approach is that it produces very long codes in order to achieve good model performance. Here are the results of IMDB sentiment analysis task:
>
> ------------------------------------------------------
>                                            code len     total size        accuracy
> GloVe baseline                     -                78 MB               87.18
> NPQ (K=60)                      480 bits       4.26 MB            87.11
> Our Model(16x32)           80 bits        1.23 MB            87.37
> ------------------------------------------------------
>
> I think it's a nice idea to separate the vector norm from quantization and it may also work in our approach to achieve higher compression rate. We will upload the revised paper once we are allowed to add revision.
>
> For Matrin's paper, their method is based on sparsification. I will try to get the codes from him or find a way to compare with his model.

---

> ### Comment · Area_Chair · 2017-12-08
> **Others**
>
> There are also several papers using sparse coding directly on word embeddings (Yogatama et al 2015?), using optimization tools like SPAMS, instead of an autoencoder. These models are not "deep" but certainly worth citing and understanding the benefits of this approach.  (Also worth comparing to See et al and Kim et al 2016? who both run pruning on the same dataset.)

---

> > ### Public Comment · (anonymous) · 2017-12-13
> > **Response to Others (Area Chair)**
> >
> > Thank you for spending the time to give us a feedback.
> >
> > Do you mean the paper [1]? I have read both [1] and [2], they are applying sparse coding to the word embeddings. Although they have a different purpose, I'm impressed by the strong interpretability they can gain through the process. I will cite the related papers in the sparse coding. As two reviewers are interesting in the interpretability of the codes, I'm also doing some experiments trying to find some ways to find the topics of learned codes.
> >
> > [1] Sparse Overcomplete Word Vector Representations (Faruqui et al., 2015)
> > [2] Learning Word Representations with Hierarchical Sparse Coding (Yogatama et al., 2014)
> >
> > For (See et al., 2016) and the compression part of (Zhang et al., 2017), we also compare with the same pruning techniques and report the results in the experiment sections. As we only compress word embeddings, we don't have the class weighting problem as discussed in (See et al., 2016). Both of the papers report a compression rate of 80% with a small performance loss, which is identical to our results. Actually, with the pruning technique, we achieved a 90% compression rate for word embeddings in the translation tasks as shown in Table 4. However, it does not work well on the sentiment analysis task.

---

### Public Comment · ~Martin_Renqiang_Min1 · 2018-01-01
**Related work missing**

Besides that this work is conceptually related to the product quantification method (https://lear.inrialpes.fr/pubs/2011/JDS11/jegou_searching_with_quantization.pdf)  mentioned by a public comment, this work is also highly related to the linear version of our recent work presented at NIPS 2017 Workshop on Machine Learning in Discrete Structures, entitled "Learning K-way D-dimensional Discrete Code For Compact Embedding Representations" (https://arxiv.org/abs/1711.03067).

We hope that the authors could mention our work in future revision of this concurrent ICLR submission. Thanks.

---

> ### Public Comment · (anonymous) · 2018-01-03
> **Response to Martin Renqiang Min**
>
> Hi, the authors of the NIPS 2017 Workshop paper have already contacted us. We share a similar idea but the work is conducted independently. They have already cited our paper in their Arxiv version. We will upload the revised version of our paper as soon as we are allowed to upload a revision. (Currently I cannot find the upload button in the revision page.)

---

### Decision · Program_Chairs · 2018-01-29
**ICLR 2018 Conference Acceptance Decision**

**Decision:**

Accept (Poster)

**Comment:**

This paper proposes an offline neural method using concrete/gumbel for learning a sparse codebook for use in NLP tasks such as sentiment analysis and MT. The method outperforms other methods using pruning and other sparse coding methods, and also produces somewhat interpretable codes. Reviewers found the paper to be simple, clear, and effective. There was particular praise for the strength of the results and the practicality of application. There were some issues, such as only being applicable to input layers, and not being able to be applied end-to-end. The author also did a very admirable job of responding to questions about analysis with clear and comprehensive additional experiments.